# Inhibitory Effect of Zinc on Colorectal Cancer by Granzyme B Transcriptional Regulation in Cytotoxic T Cells

**DOI:** 10.3390/ijms24119457

**Published:** 2023-05-29

**Authors:** Naoya Nakagawa, Yutaka Fujisawa, Huihui Xiang, Hidemitsu Kitamura, Keigo Nishida

**Affiliations:** 1Laboratory of Immune Regulation, Graduate School of Pharmaceutical Sciences, Suzuka University of Medical Science, Suzuka 513-8670, Japan; dp20001@st.suzuka-u.ac.jp; 2Faculty of Pharmaceutical Sciences, Suzuka University of Medical Science, Suzuka 513-8670, Japan; fujisawa@suzuka-u.ac.jp; 3Division of Functional Immunology, Institute for Genetic Medicine, Hokkaido University, Sapporo 060-0815, Japan; xiang@gancen.asahi.yokohama.jp (H.X.); kitamura012@toyo.jp (H.K.); 4Department of Biomedical Engineering, Faculty of Science and Engineering, Toyo University, Kawagoe 350-8585, Japan

**Keywords:** zinc, cytotoxic T cell, colorectal cancer, granzyme B, tumor immunity, calcineurin

## Abstract

Zinc is one of the essential trace elements and is involved in various functions in the body. Zinc deficiency is known to cause immune abnormalities, but the mechanism is not fully understood. Therefore, we focused our research on tumor immunity to elucidate the effect of zinc on colorectal cancer and its mechanisms. Mice were treated with azoxymethane (AOM) and dextran sodium sulfate (DSS) to develop colorectal cancer, and the relationship between zinc content in the diet and the number and area of tumors in the colon was observed. The number of tumors in the colon was significantly higher in the no-zinc-added group than in the normal zinc intake group, and about half as many in the high-zinc-intake group as in the normal-zinc-intake group. In T-cell-deficient mice, the number of tumors in the high-zinc-intake group was similar to that in the normal-zinc-intake group, suggesting that the inhibitory effect of zinc was dependent on T cells. Furthermore, we found that the amount of granzyme B transcript released by cytotoxic T cells upon antigen stimulation was significantly increased by the addition of zinc. We also showed that granzyme B transcriptional activation by zinc addition was dependent on calcineurin activity. In this study, we have shown that zinc exerts its tumor-suppressive effect by acting on cytotoxic T cells, the center of cellular immunity, and increases the transcription of granzyme B, one of the key molecules in tumor immunity.

## 1. Introduction

Zinc is one of the essential trace elements, and total zinc levels in the body are maintained at homeostasis. When zinc homeostasis is disrupted, various symptoms are known to occur. For example, zinc deficiency due to insufficient zinc intake or inhibition of zinc absorption can result in symptoms such as taste disorders, delayed wound healing, and immune abnormalities [1,2,3,4]. It has also been noted that drugs such as antimicrobials and levodopa chelate with zinc, thereby inhibiting zinc absorption from the gastrointestinal tract and contributing to drug-induced taste disorders [5,6].

Zinc is a nutrient that plays an important role in regulating immune function, and zinc deficiency is associated with a variety of diseases. Infectious diseases are among those associated with zinc deficiency. Zinc administration to children with infectious diarrhea in developing countries has been shown to shorten the duration of diarrhea [7]. In COVID-19, it was also noted that the risk of progression to severe condition is higher in adults with serum zinc levels below 70 μg/dL, and the rate of hospitalization is increased in children [8,9].

Zinc deficiency has also been associated with allergic diseases, and Kanda et al. showed that zinc induces A20, a zinc finger protein that inhibits NF-κB expression and decreases inflammatory cytokines in atopic dermatitis [10]. On the other hand, zinc and the zinc transporter ZnT5 are required for protein kinase C-mediated nuclear translocation of NF-κB in mast cells, which have been shown to be potentially involved in the development of allergic diseases [11,12]. Thus, maintaining zinc homeostasis in vivo is important in preventing zinc deficiency and the various complications associated with it.

Many studies have previously reported that zinc deficiency is involved in the progression of malignant tumors; Gyorkey et al. reported lower zinc content in the prostate tissue of prostate cancer patients compared to healthy controls [13]. Ozeki et al. also showed that low serum zinc concentration is a risk factor for hepatocellular carcinoma in patients with hepatitis C after hepatitis C virus eradication [14]. These reports suggest that low tissue and serum zinc concentrations increase the risk of malignancy, including its development and progression. Furthermore, Epstein et al. showed that higher zinc intake in prostate cancer patients decreased the risk of death from prostate cancer [15]. Thus, adequate zinc intake may reduce the risk of death from cancer.

As noted above, many reports published in the past have shown that zinc contributes to a decreased risk of cancer. However, the mechanism of the effect of zinc on tumor suppression is not fully understood. Therefore, we investigated in detail the relationship between zinc and tumor immunity.

## 2. Results

### 2.1. Increased Zinc Intake Caused a Decrease in the Frequency of Colorectal Malignancies

First, we induced malignant tumors in the colon of mice with AOM (azoxymethane) and DSS (dextran sodium sulfate) to observe the tumor-suppressive effect of zinc. The results showed that compared to mice fed a diet with 70 mg/kg of zinc (normal-zinc-intake group), the number of tumors decreased in mice fed a diet with 1000 mg/kg of zinc (high-zinc-intake group) and increased in mice with zinc content limited to less than 5 mg/kg in the diet (no-zinc-added group) (Figure 1C). Serum zinc concentrations increased in the group fed a diet with 1000 mg/kg of zinc, and no other significant changes were observed (Figure 1B). Serum copper concentrations decreased in the group fed a diet with high zinc intake, and no other significant changes were observed (Appendix A). There were no changes in average area per tumor, colon length, or mouse body weight (Figure 1D–F).

### 2.2. Removal of CD4- and CD8-Positive Cells Abolished the Difference in Tumor Frequency in the High-Zinc-Intake Group

To investigate the involvement of T cells in the tumor-suppressive effect of zinc, CD4- and CD8-positive cells were eliminated using anti-CD4 and anti-CD8 antibodies, and colorectal cancer was induced by AOM and DSS in the same manner. The results showed that the frequency of tumor development in the AOM/DSS-treated high-zinc-intake group and the AOM/DSS-treated normal-zinc-intake group was comparable (Figure 2C). Serum zinc concentration, average area per tumor, colon length, and mouse body weight were similar to those before removal of CD4- and CD8-positive cells (Figure 2A,E,G,I). There were no significant differences in serum copper (Appendix A).

### 2.3. Removal of NK1.1-Positive Cells Did Not Change the Difference in Tumor Frequency

To investigate the involvement of NK (natural killer) cells in the tumor-suppressive effect of zinc, NK1.1-positive cells were removed using anti-NK1.1 antibody and colorectal cancer induction by AOM and DSS was performed in the same manner. The results were similar to those before removal of NK1.1-positive cells (Figure 2B,D,F,H,J and Appendix A).

### 2.4. Effects of Zinc on Differentiation and Proliferation of Immunocompetent Cells

Next, the effects of zinc on the differentiation and proliferation of immunocompetent cells were examined. Flow cytometry was used to analyze changes in the differentiation and proliferation of T cells, B cells, NK cells, dendritic cells, neutrophils, and macrophages in the spleen. As a result, no effect of zinc on the differentiation and proliferation of these immunocompetent cells was observed (Figure 3B–I).

### 2.5. Effects of Zinc on the Functional Expression of Immunocompetent Cells

Since the effect of zinc on the differentiation and proliferation of immunocompetent cells was not observed, the effect of zinc on the functional expression of immunocompetent cells was analyzed. Cytokine, perforin, and granzyme B transcript levels in immunocompetent cells in response to antigen stimulation were analyzed by real-time PCR. The results showed that granzyme B transcript levels were significantly higher in the zinc-treated group compared to the control group (Figure 4E). There were no changes in IFN-γ (Interferon-γ), IL-2 (Interleukin-2), IL-4 (Interleukin-4), and perforin transcript levels (Figure 4A–D).

### 2.6. Granzyme B Is Transcriptionally Activated via the Calcineurin Pathway

To investigate the transcriptional activation pathway of granzyme B, granzyme B transcript levels in T cells in mouse spleen in response to antigen stimulation were analyzed by real-time PCR using inhibitors of signaling molecules. As a result, a marked decrease in granzyme B transcript levels was observed when CsA (Cyclosporine A) was added (Figure 5B). Granzyme B transcript levels were also decreased when wortmannin was added (Figure 5D); the addition of U0126 did not decrease granzyme B transcript levels (Figure 5C).

### 2.7. Zinc Increases the Transcriptional Activity of Granzyme B via Intracellular Signaling Molecules

To investigate whether zinc is involved in the transcriptional activation of granzyme B, granzyme B transcript levels in T cells in mouse spleens in response to antigen stimulation under various conditions were analyzed by real-time PCR. TPEN (*N*,*N*,*N*′,*N*′-Tetrakis (2-pyridylmethyl)ethylenediamine) decreased granzyme B transcript levels (Figure 5A). In addition, the direct addition of zinc to mouse splenocytes increased granzyme B transcription in a zinc-dose-dependent manner (Figure 6A). Furthermore, simultaneous addition of zinc and CsA to splenocytes decreased the amount of zinc-induced increase in granzyme B transcription (Figure 6B).

## 3. Discussion

In the present study, our data showed that zinc reduced the frequency of tumors in the colon induced by AOM and DSS. The tumor-suppressive effect of zinc on the high-zinc-intake group was not observed due to the removal of T cells, suggesting that T cells are involved in the tumor-suppressive effect of zinc. The result showing that removal of T cells increased tumors in the high-zinc-intake group is interesting, because the AOM/DSS-induced colon cancer model used in this study reported that the inhibition of inflammation suppresses tumors [16,17,18].

Zinc has been shown to induce NK cell differentiation by increasing *GATA-3* expression [19]. Therefore, to elucidate the mechanism of the tumor-suppressive effect of zinc, we investigated the effect of zinc on the differentiation and proliferation of key immunocompetent cells. In this study, we analyzed the percentages of T cells, B cells, NK cells, dendritic cells, neutrophils, and macrophages in mouse spleen, but were unable to demonstrate that zinc is involved in the differentiation and proliferation of these immunocompetent cells.

Zinc is an essential molecule when the T cell accessory receptors CD4 and CD8 associate with tyrosine kinase Lck, and has been shown to play an important role in the functional expression of T cells and other cells responsible for immunity [20]. In addition, by removing T cells from mice in this study, the tumor-suppressive effect on the high-zinc-intake group was no longer observed. Therefore, we hypothesized that zinc might be involved in the functional expression of T cells, and we observed zinc-induced functional changes of T cells in the spleen.

Granzyme B is a serine protease released by cytotoxic T cells and NK cells, and is a molecule that plays an important role in the expression of the cytotoxic activity of these cells. Granzyme B has been shown to be involved in tumor suppression by activating gasdermin E, which is involved in cellular pyroptosis [21,22], and is also known to play an important role in tumor immunity. Granzyme B is also a molecule that correlates with the anti-tumor activity of cytotoxic T cells, as PET (Positron Emission Tomography) measurements have shown that it may be used as an early biomarker of therapeutic response in cancer immunotherapy such as anti-PD-1 therapy [23]. We have shown that zinc administration promotes granzyme B transcriptional activation of T cells in the spleen.

To further investigate the mechanism by which zinc promotes granzyme B transcriptional activation, we analyzed the granzyme B transcriptional pathway. Granzyme B transcriptional activation was markedly suppressed by CsA (Figure 5B), suggesting that granzyme B transcriptional activation occurs in a calcineurin (CN)-mediated pathway, since CsA is an inhibitor of CN. The inhibition of zinc-induced granzyme B transcriptional enhancement by CsA suggests that zinc acts directly or indirectly on signaling molecules in the granzyme B transcriptional pathway to enhance granzyme B transcription. CN is also activated by a zinc-containing enzyme, superoxide dismutase (SOD), which has been shown to protect it from inactivation by reactive oxygen species [24], and zinc may maintain CN activity through SOD.

In the present study, the frequency of tumor development decreased with increasing zinc intake. Although it has been reported that low serum zinc levels are associated with risk of tumor development [14], our data showed no correlation between serum zinc levels and the development of colon cancer. In particular, the frequency of tumor incidence in the no-zinc-added group was significantly increased, despite the fact that no decrease in serum zinc levels was observed in the no-zinc-added group. In the present study, zinc could not be completely removed from the diets of the no-zinc-added group, and less than 5 mg/kg of zinc was present in the diet. Since it has been shown that restricting zinc intake increases the expression of ZIP4, which transports zinc from the intestinal tract into intestinal cells [25], it is possible that the zinc concentration in serum was maintained in the mice in the no-zinc-added group in the present study. The zinc content in the feces of the mice used in this study correlated with zinc intake but not with other metals (Appendix A). It is necessary to measure and examine zinc content not only in serum but also in various organs and body fluids.

In this study we showed that zinc may modulate signaling pathways involved in promoting granzyme B transcription. However, we have not been able to identify the molecules on which zinc acts directly. The identification of molecules on which zinc acts directly in promoting granzyme B transcription is a future challenge.

In conclusion, we have shown that zinc exerts its tumor-suppressive effect by acting on T cells, the center of cellular immunity, and increases the transcription of granzyme B, one of the key molecules in tumor immunity. We have shown that adequate intake of zinc may be protective against cancer. 

## 4. Material and Methods

### 4.1. Animals and Chemicals

ICR female mice were purchased from Japan SLC Inc. (Hamamatsu, Japan). C57BL/6 male mice were purchased from Charles River Japan (Kanagawa, Japan). This study was conducted in accordance with the recommendations of the Guide for the Care and Use of Laboratory Animals of Suzuka University of Medical Science (approval number 75). In addition, the animal experiment in the Figure 3 and Figure 4 was conducted at the animal experiment facility of Hokkaido University. The animal study protocol was approved by the Care and Use of Laboratory Animals of Hokkaido University (approval number 14-0039, 19-0035, 19-0065). All surgeries were performed under pentobarbital anesthesia and efforts were made to minimize animal suffering. Mice were acclimated for 1 week with tap water and powder diet or a pelleted diet, ad libitum, before the start of the experiment. They were housed under controlled conditions of humidity (50 ± 10%), light (12/12 h light/dark cycle), and temperature (22 ± 2 °C).

The colonic carcinogen AOM was purchased from Sigma Chemical Co. (St. Louis, MO, USA). Saline was purchased from Otsuka Pharmaceutical Co., Ltd. (Tokyo, Japan). DSS with a molecular weight of 36,000–50,000 was purchased from MP Biomedicals, Inc. (Solon, OH, USA). Zinc acetate was purchased from Nobelpharma Co., Ltd. (Tokyo, Japan). Zinc-controlled powder diet was purchased from Oriental Yeast Co., Ltd. (Tokyo, Japan). Pentobarbital was purchased from Nacalai Tesque (Kyoto, Japan). Chloroform was purchased from KANTO Chemical Co., Inc., (Tokyo, Japan).

### 4.2. Quantification of Serum Zinc Levels and Serum Copper Levels

Serum was obtained from the blood samples by centrifugation at 3000× *g* for 20 min at 4 °C and used for analysis. Serum zinc levels were measured using 5-Br-PAPS (2-(5-bromo-2-pyridylazo)-5-(N-n-propyl-N-3-sulfopropylamino)phenol) assay [26]. Serum copper levels were measured using 3,5-DiBr-PAESA (4-(3,5-dibromo-2-pyridylazo)-N-ethyl-N-sulfopropylaniline) assay [27]. Trace elements in these serum samples were measured using metalloassay kits purchased from Metallogenics (Chiba, Japan).

### 4.3. Induction of Colorectal Cancer

The experimental protocol of this study is outlined in Figure 1A. Female ICR mice were used for this experiment. The mice were quarantined for the first 7 days and then randomized according to body weight into six groups. A summary of the groupings is shown in Table 1. Group 1—the mice were intraperitoneally injected with saline. Starting 1 week before the injection, the mice were fed a powdered diet containing 70 mg/kg zinc. Group 2—the mice were intraperitoneally injected with saline. Starting 1 week before the injection, the mice were fed a powdered diet containing 1000 mg/kg zinc. Group 3—the mice were intraperitoneally injected with saline. Starting 1 week before the injection, the mice were fed a powdered diet containing < 5 mg/kg zinc. Group 4—the mice were given a single intraperitoneal injection of AOM (10 mg/kg body weight). Starting 1 week before the injection, the mice were fed a powdered diet containing 70 mg/kg zinc. Starting 1 week after the injection, the animals received 2% DSS in their drinking water for 7 days and no further treatment for 18 weeks in accordance with previously described procedures [28]. Group 5—the mice were administered with AOM/DSS as in group 4. Starting 1 week before the injection, the mice were fed a powdered diet containing 1000 mg/kg zinc. Group 6—the mice were administered with AOM/DSS as in group 4. Starting 1 week before the injection, the mice were fed a powdered diet containing < 5 mg/kg zinc. At the end of the study (week 20), all animals were checked for body weight and were sacrificed using pentobarbital. For serum preparation, blood was collected from the heart into blood collection tubes (Terumo Corporation, Tokyo, Japan) with a serum separator and coagulation promoter before the autopsy. During the autopsy, the large bowel was flushed with phosphate-buffered saline (PBS) and excised. Fecal samples were collected from the colon before washing the colon with PBS. The large bowel (from the ileocecal junction to the anal verge) was measured, dissected longitudinally along the main axis, and then washed with PBS. The tumor lesions were counted by two investigators.

From 1 week before administration of AOM, mice were administered with anti-CD4/CD8 antibodies or anti-NK1.1 antibody every week, and the experiment was conducted in the same protocol as above. Anti-CD4 antibodies, anti-CD8 antibodies, and anti-NK1.1 antibodies were purchased from SCRUM Inc. (Tokyo, Japan).

### 4.4. Mice and Cell Culture

The experimental protocol of this study is outlined in Figure 3A. Male C57BL/6 mice were randomized into two groups. Group 1—the mice were gavaged with PBS daily for 7 days. Group 2—the mice were gavaged with zinc acetate (0.3 mg/mouse) daily for 7 days. All animals were sacrificed using pentobarbital at the end of the study (Day 7). During the autopsy, the spleen was flushed with PBS and excised.

Female ICR mice were sacrificed using chloroform. During the autopsy, the spleen was flushed with PBS and excised.

### 4.5. Measurement of Cytokines

Spleen cells (5 × 10^5^ cells) were pretreated with TPEN, CsA, wortmannin, U0126, and zinc sulfate for 30 min at 37 °C. Spleen cells were stimulated with anti-CD3/CD28 antibodies for 24 h at 37 °C. After stimulation, spleen cells were collected. IL-2, IL-4, IFN-γ, perforin, and granzyme B expression were measured by real-time PCR. TPEN was purchased from Wako Pure Chemical (Osaka, Japan). CsA and wortmannin were purchased from Calbiochem (San Diego, CA, USA). U0126 was purchased from Cell Signaling Technology (Danvers, MA, USA). Anti-CD3/CD28 antibodies were purchased from BioLegend (San Diego, CA, USA).

### 4.6. Real-Time PCR

Cells were homogenized with Sepasol RNAI (Nacalai Tesque, Kyoto, Japan), and total RNA was isolated following the manufacturer’s instructions. For standard real-time PCR, cDNA was synthesized from 1 µg of total RNA with reverse transcriptase (ReverTra Ace; TOYOBO, Osaka, Japan) and 500 ng of oligo (dT) primer (Life Technologies, Grand Island, NY, USA) for 30 min at 42 °C. A portion of the cDNA was used for real-time PCR. The relative expression of *IL-2*, *IL-4*, *IFN-γ*, *perforin*, and *granzyme B* gene was determined compared to a reference gene *g3pdh* using the SYBR^®^ Green reagent (TaKaRa Bio Inc., Kusatsu, Japan). Primers used in these experiments were purchased from TaKaRa, Fasmac Co., Ltd. (Kanagawa, Japan) or Thermo Fisher Scientific K.K. (Tokyo, Japan), and the sequences of primers were as shown in Table 2.

### 4.7. Flow Cytometry

T cells, B cells, NK cells, dendritic cells, neutrophils, and macrophages in the spleen were measured by flow cytometry. The fluorescent signals were detected using a FACSCanto II™ flow cytometer (BD Biosciences, San Diego, CA, USA).

### 4.8. Mineralization of Murine Fecal Samples

Feces were collected immediately after defecation, transferred to cryotubes, and stored frozen. Murine fecal samples were stored at −80 °C until mineralization treatment. A pellet of each fecal sample was transferred to a 2 mL polypropylene tube, followed by lyophilizing overnight to remove excess moisture from the sample. Two quality control (QC) samples, which were prepared by loading each element in 2 mL polypropylene tubes at the amount shown in Table 3, were also lyophilized overnight. The dried fecal samples were weighed and digested using 0.5 mL of nitric acid Ultrapur^TM^-100 (KANTO Chemical Co., Inc., Tokyo, Japan) according to the following protocol [29,30]. The nitric-acid-soaked samples were pretreated by sonicating at 50 °C for 2 h and then stored at room temperature overnight. Secondly, the samples were heated to 60 °C on the dry bath and the temperature was held for 1 h. Eventually, the samples were completely digested at 90 °C for further 6 h, and allowed to cool to room temperature. Three blank tubes and two QC samples were also treated by the same digestion processes. Next, 1 mL of milliQ water was added to each treated tube, and then the diluted sample was passed through a 0.45 μm PTFE membrane filter. The tube and membrane filter were thoroughly rinsed with another 1.5 mL of milliQ water three times, and all filtrates were combined in 15 mL polypropylene tubes. Finally, the total volume was adjusted to 10 mL with milliQ water to prepare the analytical sample at 3.5% final nitric acid concentration. Furthermore, analytical samples, except for blank and QC samples, were diluted to one-fortieth with 3.5% nitric acid.

### 4.9. Quantification of Trace Metals in Murine Fecal Samples

Each element concentration in blank samples, QC samples, 1/40-diluted, or non-diluted analytical samples was analyzed with inductively coupled plasma mass spectrometry (ICP-MS) using an Agilent 7500cx equipped with an autosampler (Santa Clara, CA, USA). A multielement mixed standard solution XSTC-469 was purchased from SPEX (Metuchen, NJ, USA). The single-element standard solutions of cobalt, gallium, or yttrium were purchased from KANTO Chemical Co., Inc. (Tokyo, Japan). The external calibration standards were prepared and analyzed at the concentration range as described in Table 3. The solution containing 1 ppm of gallium and yttrium was used as an internal standard (IS), and constantly injected into ICP-MS by a perimeter pump at one-twentieth of the sample flow rate via a different line from the analytical samples. For quantitative analysis of chromium, iron, zinc, copper, and molybdenum in the analytical samples, high-purity helium gas (99.999 vol%) was used in collision gas mode. In the case of manganese, cobalt and selenium, high-purity hydrogen gas (99.99999 vol%) was used in reaction gas mode. For the analysis of ten elements, the following isotopes were monitored by ICP-MS: ^53^Cr, ^55^Mn, ^57^Fe, ^59^Co, ^63^Cu, ^66^Zn, ^71^Ga, ^78^Se, ^89^Y, and ^95^Mo. Gallium was used as IS for quantification of Cr, Mn, Fe, Co, Cu, and Zn, while yttrium was used for Se and Mo. The element concentration in the analytical sample was converted to the element levels in murine fecal sample using the following formula:$$\mathrm{element}\;\mathrm{levels}\;\mathrm{in}\;\mathrm{dried}\;\mathrm{fecal}\;\mathrm{sample}\;\left(\mu g/g\;\mathrm{of}\;\mathrm{fecal}\;\mathrm{sample}\right)\;=\frac{\left(\mathrm{concentration}\;\mathrm{in}\;\mathrm{analytical}\;\mathrm{sample}\times \mathrm{dilution}\;\mathrm{factor}-\mathrm{mean}\;\mathrm{concentration}\;\mathrm{in}\;\mathrm{blank}\;\mathrm{sample}\right)\times 10}{\mathrm{dry}\;\mathrm{weight}\;\mathrm{of}\;\mathrm{fecal}\;\mathrm{sample}}$$

For calculation of Cr, Fe, Co, Cu, Se, and Mo, the dilution factor was 1. For calculation of Mn and Zn, the dilution factor was 40.

### 4.10. Statistical Analysis

All data were statistically analyzed using Tukey’s HSD test or Student’s two-tailed *t* test with IBM SPSS Statistics software (version 26) and Microsoft Excel (2019). Data were considered statistically significant when the *p*-value was less than 0.05.

## Figures and Tables

**Figure 1 ijms-24-09457-f001:**
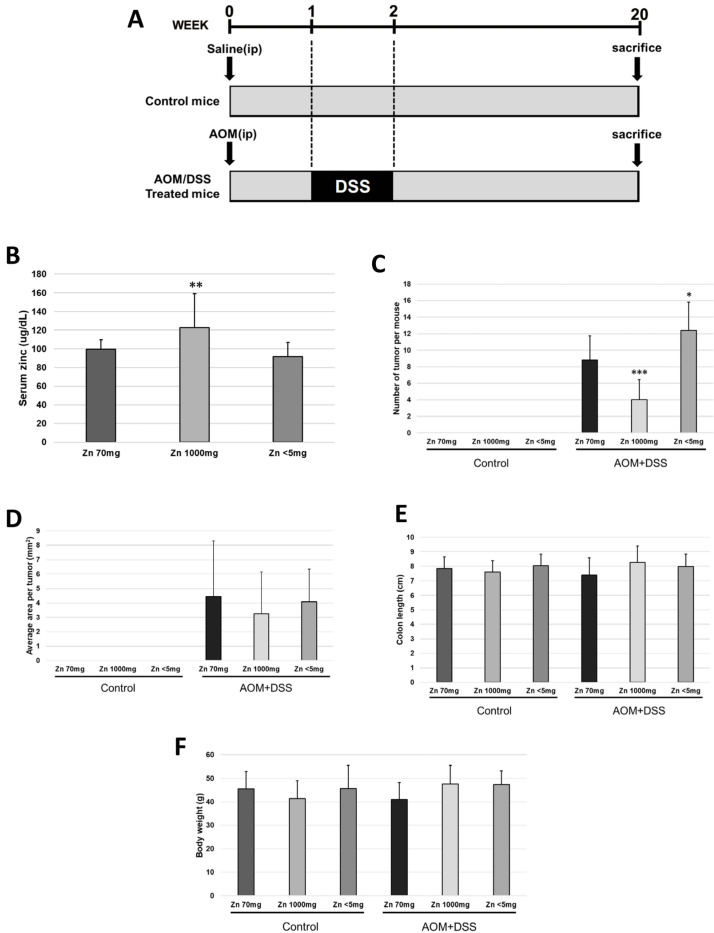
Effect of zinc administration on colon cancer induced by azoxymethane (AOM) and dextran sodium sulfate (DSS). Female ICR mice (control mice, Zn = 70 mg n = 13, Zn = 1000 mg n = 12, Zn < 5 mg n = 7; AOM/DSS-administered mice, Zn = 70 mg n = 16, Zn = 1000 mg n = 11, Zn < 5 mg n = 10) were used for this experiment. (**A**) Experimental protocol; (**B**) serum zinc level collected at week 20; (**C**) number of tumors per mouse; (**D**) average area per tumor; (**E**) colon length; and (**F**) body weight measured at week 20 (bar: SD; * *p* < 0.05; ** *p* < 0.01; *** *p* < 0.001; Tukey’s HSD test).

**Figure 2 ijms-24-09457-f002:**
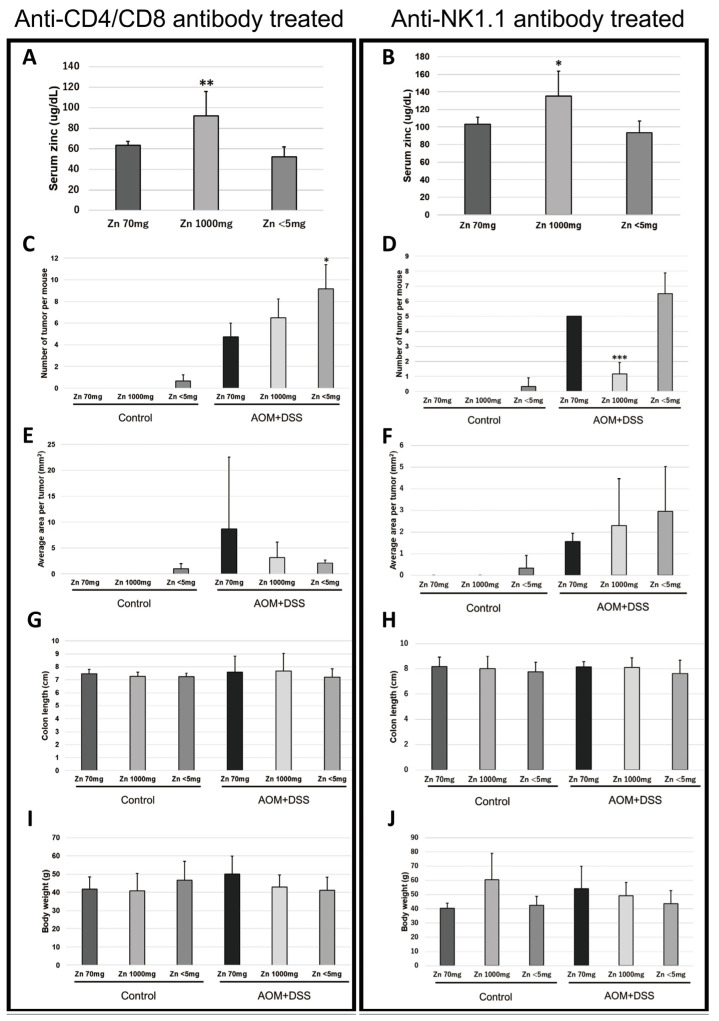
Effect of anti-CD4/CD8 antibody and zinc administration or anti-NK1.1 antibody and zinc administration on colon cancer induced by azoxymethane (AOM) and dextran sodium sulfate (DSS). Female ICR mice (anti-CD4/CD8-antibody-administered control mice, Zn = 70 mg n = 2, Zn = 1000 mg n = 3, Zn < 5 mg n = 3; anti-CD4/CD8 antibody- and AOM/DSS-administered mice, Zn = 70 mg n = 4, Zn = 1000 mg n = 4, Zn < 5 mg n = 6; anti-NK1.1-antibody-administered control mice, Zn = 70 mg n = 3, Zn = 1000 mg n = 2, Zn < 5 mg n = 3; anti-NK1.1-antibody- and AOM/DSS-administered mice, Zn = 70 mg n = 4, Zn = 1000 mg n = 6, Zn < 5 mg n = 6) were used for this experiment. (**A**) Serum zinc level collected at week 20 (anti-CD4/CD8 antibody administration); (**B**) serum zinc level collected at week 20 (anti-NK1.1 antibody administration); (**C**) number of tumors per mouse (anti-CD4/CD8 antibody administration); (**D**) number of tumors per mouse (anti-NK1.1 antibody administration); (**E**) average area per tumor (anti-CD4/CD8 antibody administration); (**F**) average area per tumor (anti-NK1.1 antibody administration); (**G**) colon length (anti-CD4/CD8 antibody administration); (**H**) colon length (anti NK1.1 antibody administration); (**I**) body weight measured at week 20 (anti-CD4/CD8 antibody administration); and (**J**) body weight measured at week 20 (anti-NK1.1 antibody administration). (bar: SD; * *p* < 0.05; ** *p* < 0.01; *** *p* < 0.001; Tukey’s HSD test).

**Figure 3 ijms-24-09457-f003:**
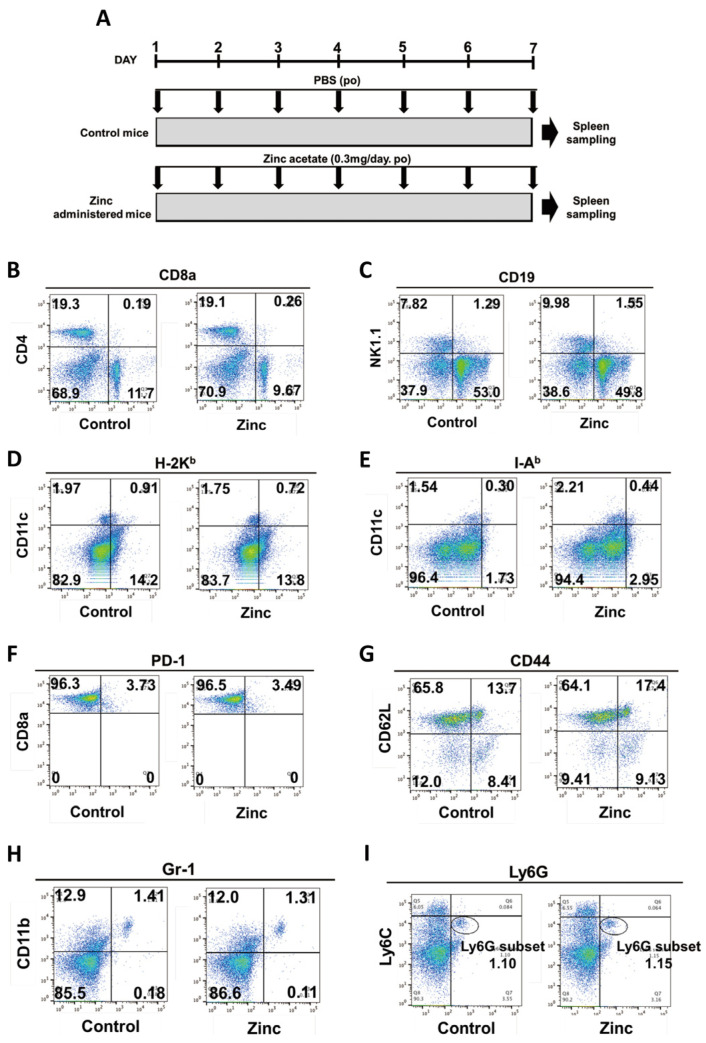
Effects of zinc administration on differentiation and proliferation of immunocompetent cells. Male C57BL/6 mice (n = 4, each) were used for this experiment. The color of the dots indicates the density of the cells. Representative profiles and the percentage of cell numbers in each quadrant are indicated in the figure. (**A**) Experimental protocol; (**B**) FACS plots marked by marker CD4 and marker CD8a; (**C**) FACS plots marked by marker NK1.1 and marker CD19; (**D**) FACS plots marked by marker CD11c and marker H-2K^b^; (**E**) FACS plots marked by marker CD11c and marker I-A^b^; (**F**) FACS plots marked by marker CD8a and marker PD-1; (**G**) FACS plots marked by marker CD62L and marker CD44; (**H**) FACS plots marked by marker CD11b and marker Gr-1; and (**I**) FACS plots marked by marker Ly6C and marker Ly6G. Ly6G subsets were circled and percentage of cells was indicated. These data were analyzed using two-tailed Student’s *t*-test, which were not significantly different.

**Figure 4 ijms-24-09457-f004:**
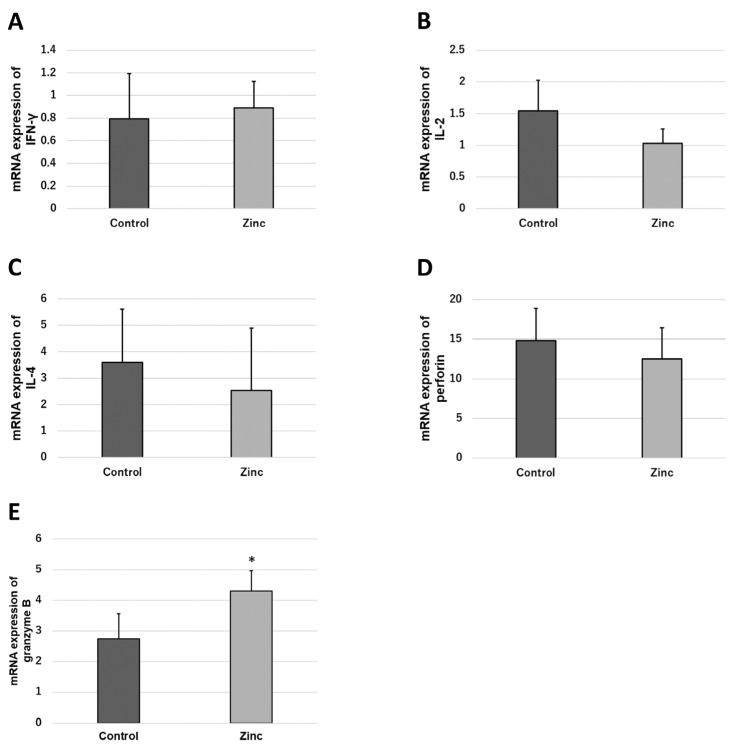
Effect of zinc administration on functional expression of T-cells. Male C57BL/6 mice (n = 4, each) were used for this experiment. Spleen cells were antigen-stimulated by anti-CD3/CD28 antibody (50 ng/mL). (**A**) mRNA expression of interferon-γ (IFN-γ); (**B**) mRNA expression of interleukin-2(IL-2); (**C**) mRNA expression of interleukin-4(IL-4); (**D**) mRNA expression of perforin; and (**E**) mRNA expression of granzyme B. (bar: SD; * *p* < 0.05; two-tailed Student’s *t*-test).

**Figure 5 ijms-24-09457-f005:**
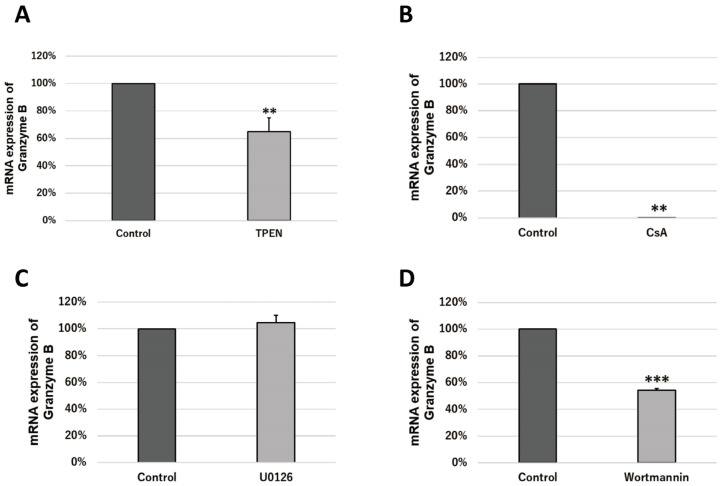
Effect of zinc chelator and effect of inhibitors on signaling molecules on granzyme B transcription of T-cells. Female ICR mice were used for this experiment. Spleen cells were antigen-stimulated by anti-CD3/CD28 antibody (50 ng/mL). An expression level of 100% indicates no effect of the inhibitor. These experiments were performed at least three times independently, and representative data was shown. (**A**) mRNA expression of granzyme B when 2.5 μM *N*,*N*,*N*′,*N*′-(2-pyridylmethyl)ethylenediamine (TPEN) was added; (**B**) mRNA expression of granzyme B when 0.1 μM ciclosporin A(CsA) was added; (**C**) mRNA expression of granzyme B when 0.1 μM U0126 was added; and (**D**) mRNA expression of granzyme B when 0.1 μM wortmannin was added. (bar: SD; ** *p* < 0.01; *** *p* < 0.001; two-tailed Student’s *t*-test).

**Figure 6 ijms-24-09457-f006:**
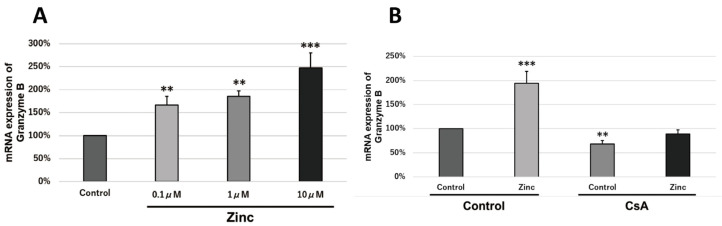
Calcineurin-activity-dependent activation of granzyme B transcription by zinc. Female ICR mice were used for this experiment. An expression level of 100% indicates no effect of the inhibitor and zinc. These experiments were performed at least three times independently, and representative data were shown. (**A**) mRNA expression of granzyme B when zinc sulfate was added. Spleen cells were antigen-stimulated by anti-CD3/CD28 antibody (10 ng/mL); (**B**) mRNA expression of granzyme B when 10 nM CsA and 1 μM zinc sulfate was added. Spleen cells were antigen-stimulated by anti-CD3/CD28 antibody (50 ng/mL). (bar: SD; ** *p* < 0.01; *** *p* < 0.001; Tukey’s HSD test).

**Table 1 ijms-24-09457-t001:** A summary of the groupings for AOM/DSS-induced colorectal cancer model and feed condition.

	AOM/DSS Treatment	Diet
Group 1	Untreated	70 mg/kg zinc
Group 2	Untreated	1000 mg/kg zinc
Group 3	Untreated	<5 mg/kg zinc
Group 4	Treated	70 mg/kg zinc
Group 5	Treated	1000 mg/kg zinc
Group 6	Treated	<5 mg/kg zinc

**Table 2 ijms-24-09457-t002:** The sequences of primers.

Target	Primer	Sequence
IL-2	Forward Primer	5′-CCTGAGCAGGATGGAGAATTACA-3′
	Reverse Primer	5′-TCCAGAGACATGCCGCAGAG-3′
IL-4	Forward Primer	5′-TCTCGAATGTACCAGGAGCCATATC-3′
	Reverse Primer	5′-AGCACCTTGGAAGCCCTACAGA-3′
IFN-γ	Forward Primer	5′-CGGCACAGTCATTGAAAGCCTA-3′
	Reverse Primer	5′-GTTGCTGATGGCCTGATTGTC-3′
Perforin	Forward Primer	5′-GCAATTTCCGGGCAGAACA-3′
	Reverse Primer	5′-CTGAACTCCTGGCCACCAAAG-3′
Granzyme B	Forward Primer	5′-GACTTTGTGCTGACTGCTGCTC-3′
	Reverse Primer	5′-GGGATGACTTGCTGGGTCTTC-3′
G3PDH	Forward Primer	5′-AGCTGAACGGGAAGCTCACT-3′
	Reverse Primer	5′-TGAAGTCGCAGGAGACAACC-3′

**Table 3 ijms-24-09457-t003:** The quantitation range of each element and accuracy assurance on ICP-MS analysis.

Monitored Isotopes	Concentration Range in Standards (ppb)	Concentration Range in Analytical Samples (ppb)	Amount in QC Sample (ng) *	Mean Recovery (%) **
^53^Cr	0.750–250	3.60–21.69	225	77.1
^55^Mn	1.243–124.25	1.74–7.06	1118.25	101.9
^57^Fe	3.72–1240	125.2–1158	1116	97.2
^59^Co	0.025–25.0	0.039–0.261	22.5	107.8
^63^Cu	0.750–250	27.90–177.4	225	92.7
^66^Zn	3.713–1237.5	4.52–761.0	1113.75	125.6
^78^Se	0.249–74.7	(0.236) ***, 0.256–3.40	224.1	100.6
^95^Mo	0.074–24.78	0.131–1.41	222.975	95.5

* The values present the loading amount of each element in the 2 mL polypropylene tube. ** The values indicate the mean recovery of each element in QC samples with the digestion protocol (n = 2). *** The parenthesized value, which was under a low limit of quantitation in only one analytical sample, allowed us to calculate the element levels in fecal sample by extrapolation.

## Data Availability

Not applicable.

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
