# Peer review of "Inhibitory Effect of Zinc on Colorectal Cancer by Granzyme B Transcriptional Regulation in Cytotoxic T Cells"

_ijms, 2023, doi:10.3390/ijms24119457_

Round 1

Reviewer 1 Report

The manuscript described the study on the intake of Zinc to reduce colon cancer incidence. Basically, the research is interesting and after consideration of making the following revisions, the reviewer supports the publication of the manuscript.

1) the resolution of all the text in the figures is not good enough (text blurred) and should be improved

2) Figure 3 the image of flow cytometry is better to be colorful 

3) granzyme B is important in this study, so can the author provide protein expressions of granzyme B such as western blot or IHC images? This can better support the author's viewpoint. 

The writing of manuscript is good. 

Reviewer 2 Report

The manuscript addresses an important and relevant topic regarding the impact of minerals and micronutrients on the occurrence and evolution of cancer. Even that is a study on animal I think is relevant for the topic, considering that it is in vivo study.   The importance of diet micronutrients in the evolution of cancer is high, especially if we consider future studies on the prevention of colorectal cancer.   It looks like the experiment proves a direct relationship between the aggressivity of colon cancer and zinc intake, not necessarily via immune modulation. I consider it an original research.   I am not able to judge the quality of the material used for the experiments but they are very well described.   I think that the authors should add a paragraph for conclusions, which is not present in the manuscript. There is just one sentence.   Some references are really old, but there are also some new articles cited, mostly on COVID patients.   All tables and figures reflect the experiment results and I consider they are well explained.
